# Cross-Reactivity of Antibodies in Intravenous Immunoglobulin Preparation for Protection against SARS-CoV-2

**DOI:** 10.3390/microorganisms11020471

**Published:** 2023-02-13

**Authors:** Toshifumi Osaka, Yoko Yamamoto, Takehisa Soma, Naoko Yanagisawa, Satoru Nagata

**Affiliations:** 1Department of Microbiology and Immunology, Tokyo Women’s Medical University, Tokyo 162-8666, Japan; 2Department of Pediatrics, Tokyo Women’s Medical University, Tokyo 162-8666, Japan; 3Veterinary Diagnostic Laboratory, Marupi Lifetech Co., Ltd., Osaka 563-0011, Japan

**Keywords:** SARS-CoV-2, immunoglobulin, cross-reactive antibody, pseudovirus entry assay, antibody-dependent enhancement

## Abstract

Severe cases of COVID-19 continue to put pressure on medical operations by prolonging hospitalization, occupying intensive care beds, and forcing medical personnel to undergo harsh labor. The eradication of SARS-CoV-2 through vaccine development has yet to be achieved, mainly due to the appearance of multiple mutant-incorporating strains. The present study explored the utility of human intravenous immunoglobulin (IVIG) preparations in suppressing the aggravation of any COVID-19 infection using a SARS-CoV-2 pseudovirus assay. Our study revealed the existence of IgG antibodies in human IVIG preparations, which recognized the spike protein of SARS-CoV-2. Remarkably, the pretreatment of ACE2/TMPRSS2-expressing host cells (HEK293T cells) with IVIG preparations (10 mg/mL) inhibited approximately 40% entry of SARS-CoV-2 pseudovirus even at extremely low concentrations of IgG (0.16–1.25 mg/mL). In contrast, the antibody-dependent enhancement of viral entry was confirmed when SARS-CoV-2 pseudovirus was treated with some products at an IgG concentration of 10 mg/mL. Our data suggest that IVIG may contribute to therapy for COVID-19, including for cases caused by SARS-CoV-2 variants, since IVIG binds not only to the spike proteins of the virus, but also to human ACE2/TMPRSS2. An even better preventive effect can be expected with blood collected after the start of the COVID-19 pandemic.

## 1. Introduction

An outbreak of coronavirus infectious disease 2019 (COVID-19), caused by severe acute respiratory syndrome coronavirus 2 (SARS-CoV-2), occurred in 2019, and the infection spread worldwide in a short period of time [1]. Some patients with COVID-19 develop severe pneumonia or acute respiratory distress syndrome (ARDS) [2]. Although the details concerning the mechanism underlying COVID-19 aggravation have not yet been elucidated, an abnormal immune response enhancement (e.g., increased production of inflammatory cytokines, activation of the complement system) was reportedly found in patients with severe symptoms [3].

At present, a vaccine-induced humoral-immune response and passive immunization with the convalescent serum of COVID-19 patients and human monoclonal antibodies are used worldwide as effective prophylaxis and therapy against SARS-CoV-2 infection [4,5,6,7,8,9,10]. However, viral replication in the presence of neutralizing antibodies against SARS-CoV-2 has raised concerns about the emergence of neutralization escape in variants with increased transmissibility and reduced vaccine effectiveness because of numerous mutations in the spike protein [11].

Human intravenous immunoglobulin (IVIG) preparations have been proven to have suitable clinical safety and efficacy in terms of the prevention and treatment of various infectious diseases and autoimmune diseases (e.g., idiopathic thrombocytopenic purpura, Kawasaki disease, Guillain-Barré syndrome) [12]. Possible molecular mechanisms underlying the effects of IVIG treatment include Fc receptor (FcR) blocking, the cross-protection of various pathogens, regulation of the cytokine production in immune cells (e.g., lymphocytes and monocytes), and the suppression of complement system activation [13]. However, while the utility of IVIG preparations for ARDS or severe pneumonia caused by COVID-19 has been examined, scientific evidence supporting its efficacy in cases of SARS-CoV-2 infection has not yet been established [14,15,16,17,18,19]. Human IVIG preparations may contain cross-reactive antibodies against the spike protein of SARS-CoV-2 or virus entry-mediated host molecules, such as human angiotensin-converting enzyme 2 (ACE2) receptor and transmembrane serine protease 2 (TMPRSS2) [20]. The present study therefore examined whether human IVIG preparations are able to protect host cells against SARS-CoV-2 infection using a pseudovirus entry assay.

## 2. Materials and Methods

### 2.1. IVIG Preparation

We used four kinds of commercially available IVIG preparations: freeze-dried sulphonated human normal immunoglobulin (Product-A), pH 4-treated acidic human normal immunoglobulin (product-B), and polyethylene glycol-treated human normal immunoglobulin (products C and D) [21]. The IVIG preparations used in this study were manufactured from a healthy Japanese donor before the COVID-19 outbreak in Japan (before 1 March 2020).

### 2.2. Evaluating the SARS-CoV-2-Derived Antigen-Binding Ability of the IVIG Preparation

The spike protein derived from SARS-CoV-2 was immobilized on a plate, and the antigen-binding ability of the IVIG preparation was measured by an enzyme-linked immunosorbent assay (ELISA). Wild-type trimeric SARS-CoV-2 Spike Antigen (5 µg; BioServUK, Ltd., Sheffield, UK) was immobilized on a Corning^®^ 96-well clear flat-bottom polystyrene high-bind microplate (Corning, NY, USA), and blocking using 10% fetal bovine serum (FBS)-phosphate-buffered saline (PBS) solution was carried out for 30 min at 37 °C. After washing with PBS-T solution (0.1% Tween 20), the serially diluted IVIG preparations were incubated for 1 h at 37 °C. After washing with PBS-T, the plate was incubated with a 1:4000 dilution of horseradish peroxidase (HRP)-labeled anti-human IgG antibody goat Anti-Human IgG ab6858 (Abcam, Boston, MA, USA) as the detection antibody, for 30 min at 37 °C. As a positive control, a serially diluted anti-SARS-CoV-2 spike protein antibody CR3022 (Abcam) was used. After washing, the wells were incubated with tetramethylbenzidine (TMB) substrate (Agilent Technology, Santa Clara, CA, USA). An acidic stopping solution was added, and the absorbance at 450 nm was measured.

Furthermore, the presence of neutralizing IgG antibodies in the commercially available IVIG preparations was verified using the SARS-CoV-2 Anti-RBD Antibody Profiling Kit (MBL Co., Ltd., Tokyo, Japan), as briefly mentioned below. A recombinant His-tagged RBD protein was incubated with a neutralizing antibody (as a positive control) or with serially diluted IVIG preparations for 30 min at room temperature. The mixtures were then added to a 96-well microplate immobilized with ACE2 for 30 min at room temperature. After washing, HRP-conjugated anti-His-tagged monoclonal antibody was added to the wells and then incubated for 30 min at room temperature. After washing, TMB substrate solution was added to the wells and then incubated for 15 min at room temperature. After the addition of acidic stopping solution, the absorbance at 450 nm was measured. The inhibition rate of each test sample was calculated using the following formula: inhibition rate (%) = {1 − (absorbance value of sample) / (absorbance value of blank)} × 100.

### 2.3. Evaluating the Infection-Preventive Effect of IVIG Preparations Using a Pseudovirus Entry Assay

SARS-CoV-2 spike protein-pseudotyped lentivirus was prepared using Lenti-X SARS-CoV-2 Packaging Single Shots (wild type, full length) (Takara Bio, Shiga, Japan), a pLVSX-luciferase-puro vector (Takara Bio), and HEK293T-LentiX cells (Takara Bio). Culture cells for the assay were cultured with Dulbecco’s modified eagle medium (DMEM) containing 10% tetracycline-free FBS (Takara Bio) and penicillin-streptomycin solution (Fujifilm Wako Pure Chemical, Ltd., Osaka, Japan). The SARS-CoV-2 pseudovirus titer in the culture supernatants filtered with a 0.45 μm cellulose acetate filter was determined using a Lenti-XTM p24 Rapid Titer Kit (Takara Bio). HEK293T cells stably expressing human ACE2 and TMPRSS2 (GeneCopoeia, Rockville, MD, USA) were used for a pseudovirus entry assay. ACE2/TMPRSS2-expressing HEK293T cells (1 × 10^5^ cells) were seeded in a white-walled 96-well microplate (Nunc MicroWell 96-Well, Nunclon Delta-Treated, Flat-Bottom Microplate; Thermo Fisher Scientific, Paisley, UK), and then incubated in a 5% CO_2_ incubator (37 °C, 24 h).

For the evaluation of the infection-preventive effect of IVIG preparations through its binding to cellular surface molecules, ACE2/TMPRSS2-expressing HEK293T cells were reacted with IVIG preparations (100 µL, 10 mg-IgG/mL) at 37 °C for 1 h before infection with the SARS-CoV-2 pseudovirus. After the removal of IVIG preparations from each well, 4 two-fold dilutions of the SARS-CoV-2 pseudovirus (multiplicity of infection: MOI = 25, 50, 100, 200) were administered to the ACE2/TMPRSS2-expressing HEK293T cells pre-treated with the IVIG preparations. To characterize the SARS-CoV-2 pseudovirus neutralization of the IVIG preparation, 1 × 10^7^ lentivirus particles (LP) of the SARS-CoV-2 pseudovirus (MOI = 100) were reacted with 7 two-fold serial dilutions of IVIG preparations (0.156 to 10 mg-IgG/mL) at 37 °C for 1 h, then added to each well of ACE2/TMPRSS2-expressing HEK293T cells treated with or without IVIG preparations. After the plates were incubated at 37 °C for 24 h in a 5% CO_2_ incubator, the firefly luciferase activity was determined using a One-Glo assay system (Promega, San Luis Obispo, CA, USA) on a GloMax^®^ Discover Microplate Reader (Promega).

All assays were conducted in three independent experiments in hexaplicate. After subtracting the background (average RLU in negative control wells), the inhibition rate of the IVIG test wells was calculated using the following formula: inhibition rate (%) = {1 − (RLU of IVIG-test well)/(average RLU of positive control well)} × 100.

## 3. Results

### 3.1. Human IVIG Preparations Contain IgG Antibodies against the Spike Protein of SARS-CoV-2

An ELISA using a full-length trimeric spike protein of wild-type SARS-CoV-2 was performed to determine the presence of IgG antibodies binding to the spike protein of SARS-CoV-2 in IVIG preparations. Antibody CR3022, which was screened as a neutralizing antibody against SARS-CoV-1, was found to bind to the spike protein of SARS-CoV-2 (Figure 1A) [22]. This result is attributed to high conservation in the receptor binding domains (RBDs) between SARS-CoV-1 and SARS-CoV-2 [23]. In addition, it was confirmed that all IVIG preparations tested in this study contained cross-reactive IgG antibodies against the spike protein of wild-type SARS-CoV-2 (Figure 1A).

Next, we determined whether neutralizing antibodies were involved in the IVIG preparations. A neutralizing antibody as a positive control inhibited the binding of recombinant RBD protein to ACE2 (Figure 1B). These results suggest that the IVIG preparations contain IgG antibodies that recognize the spike protein of SARS-CoV-2, but not neutralizing antibodies.

### 3.2. A Potential Preventive Effect of IVIG Preparations for SARS-CoV-2 Infection

We investigated whether IVIG preparation can prevent SARS-CoV-2 infection using a pseudovirus entry assay. A linear correlation (R^2^ = 0.9994) was observed between the SARS-CoV-2 pseudovirus load and the relative light units (RLU) on ACE2/TMPRSS2-expressing HEK293T cells (Figure 2A). The pretreatment of ACE2/TMPRSS2-expressing HEK293T cells with IVIG preparations (IgG concentration, 10 mg/mL) was found to inhibit pseudovirus infection at a rate of approximately 10–40% (Figure 2B), suggesting that cross-reactive IgG antibodies in IVIG preparations interacted with cell surface molecules involved in the cellular internalization of SARS-CoV-2. In addition, we observed the inhibition of pseudovirus entry upon the treatment of SARS-CoV-2 pseudovirus with IVIG preparations, even at low concentrations (Figure 2C, left panel). For example, at an IgG concentration of 1.25 mg/mL, the inhibition rates for products A and C were 29% and 20%, respectively. Furthermore, the pretreatment of host cells with all IVIG preparations (10 mg/mL) resulted in approximately 30% to 40% inhibition of cell entry of SARS-CoV-2 pseudovirus that had been reacted with IVIG preparations at IgG concentrations of 0.16–2.5 mg/mL (Figure 2C, right panel).

In contrast, the antibody-dependent enhancement (ADE) of viral entry was confirmed when SARS-CoV-2 pseudovirus was treated with product B at an IgG concentration of 10 mg/mL (Figure 2C, left panel). Furthermore, the pretreatment of ACE2/TMPRSS2-expressing HEK293T cells with product B failed to inhibit the infectivity enhancement of SARS-CoV-2 pseudovirus that had been reacted with product B (10 mg/mL) (Figure 2C, right panel).

## 4. Discussion

While neutralizing antibody-mediated therapies (e.g., vaccination, monoclonal antibody cocktails) have played an important role in the protection against SARS-CoV-2, it is crucial to establish a treatment method for patients with severe symptoms due to the epidemic of SARS-CoV-2 variants [24]. Remarkably, our data showed that the effect of IVIG in preventing the infection of SARS-CoV-2 pseudovirus was attributed more to the blocking of cell surface molecules (e.g., ACE2, TMPRSS2) than to that of the SARS-CoV-2 spike protein, suggesting that IVIG preparations may confer a preventive effect against SARS-CoV-2 infection, regardless of variants.

All IVIG preparations tested in this study were found to exert a preventive effect against SARS-CoV-2 pseudovirus infection, even at low concentrations (e.g., IgG concentration of 1.25 mg/mL). This corresponded to approximately 6 g of IVIG preparation (one or two vials) for a 70 kg adult, considering the circulating plasma volume. Low-dose IVIG treatment is, thus, feasible as a substitution for subcutaneous administration, and its safety has already been proven in outpatient cases [25]. Recently, it has also been reported that low-dose IVIG treatment in patients with sepsis results in a significant improvement in disseminated intravascular coagulation (DIC) and platelet counts [26]. DIC and thrombosis have also been described as complications in severe COVID-19 patients with a dysregulated immune response [27]. Thus, IVIG preparations are likely to be promising existing drugs for the treatment of coagulopathy and DIC before fatal complications arise in COVID-19.

IVIG preparations in the present study contained IgG antibodies with cross-reactivity against the SARS-CoV-2 spike protein but a weak neutralizing effect. Recent studies have shown that not only neutralizing antibodies, but also non-neutralizing antibodies play a crucial role in the protection against some viral infections (e.g., human immunodeficiency virus 1, influenza virus, Marburg virus, SARS-CoV-2) through Fc-mediated effector functions, such as antibody-dependent cellular cytotoxicity (ADCC) and antibody-dependent cellular phagocytosis (ADCP) [28,29,30,31]. Fc-mediated effector functions are known to be regulated by the structure of antibodies (e.g., glycosylation, subclass) [32]. It has recently been shown that COVID-19 severity is associated with higher ADCC activity and lower ADCP activity [33], which may suggest the involvement of another preventive mechanism of IVIG by inhibiting excess inflammation through its nonspecific binding to the FcRs of macrophages, in addition to the neutralizing effect of IVIG towards SARS-CoV-2 and the antibody-dependent cell-mediated cytotoxicity towards them via FcRs. COVID-19 patients often complain of long-term sequelae in various organs, now referred to as post-COVID-19 syndrome or long COVID [34]. Patients with post-COVID syndrome demonstrate increased levels of inflammatory cytokines via the persistent activation of mononuclear phagocytes (e.g., macrophages, plasmacytoid dendritic cells) [35]. Treatment with IVIG has been applied to several inflammatory disorders, as IVIG confers anti-inflammatory and immuno-modulatory effects [36]. Thus, IVIG may also be expected to contribute to the prevention of post-COVID syndrome by promoting convergence of the inflammatory response in COVID-19. Further studies are needed to understand the impact of IVIG treatment on Fc-mediated effector functions in COVID-19.

In the family Coronaviridae, the FcR-mediated internalization of the virus and conformational changes in the RBD of the spike protein have been reported as molecular mechanisms of ADE [37,38]. This phenomenon might explain the recent findings of respiratory and neurological sequelae of mild COVID-19 in subacute stages when hosts produce various antibodies to SARS-CoV-2 [34], since recurrent COVID-19 infections may increase non-anti-RBD antibodies, which can infect a combination of host cells, possibly leading to the expansion of inflammation. The ADE induced by product B treatment at a high concentration might be attributed to an FcR-independent mechanism, as HEK293T cells did not endogenously express any FcR [38]. Although antibodies against a specific domain of the N-terminal domain (NTD) of the SARS-CoV-2 spike protein in the serum of COVID-19 patients exert an ACE2-binding enhancement effect by inducing conformational changes in the RBD, anti-NTD antibodies are usually undetectable or present at low levels in the sera of uninfected individuals [37]. Thus, the enhanced infection of viruses highly bound with cross-reactive IgG antibodies in the IVIG preparation of product B is unlikely to be attributable to anti-NTD antibodies. The ADE of the SARS-CoV-2 pseudovirus entry by the IVIG treatment was specific to product B, and a similar trend was confirmed in other lots of product B. The manufacturing process of product B involves liquid incubation treatment under a low-pH condition (approximately 4) to inactivate the virus, stabilize the IgG molecule, and increase the IgG monomer content. A low pH has been reported to increase the infectivity of SARS-CoV-2 by strengthening the binding between ACE2 and RBDs [39]. As a result, the disappearance of ADE and the preventive effect of viral infection by increasing the dilution rate of product B with DMEM medium were observed (Figure 2C), suggesting that the ADE of pseudovirus entry may be due to a decrease in the pH. Taken together, our data suggest the possible protective effect of a commercially available IVIG preparation in the development and exacerbation of COVID-19 through binding of the SARS-CoV-2 spike protein and host cell membrane proteins involved in viral infection.

One limitation associated with this study is that our data may be attributed to the use of IVIG preparations derived from healthy Japanese residents, whose fatality rate due to COVID-19 is currently about one-tenth of the world average [40]. A cohort study in Canada (Centre Hospitalier De L’Université De Montréal [CHUM]) reported no immunoglobulins against the spike proteins of SARS-CoV-2 in the plasma samples of uninfected individuals before vaccination [41]. The IVIG preparations used in the present study were processed from blood donated before significant community-based transmission of COVID-19 became apparent in Japan, which makes it possible to speculate that an even better inhibitory effect can be expected when using IVIG preparations from blood collected after the COVID-19 pandemic and vaccination campaign began. Therefore, the effectiveness of IVIG using Japanese blood in the pre-pandemic term may be much lower than what was fundamentally intended.

Although the development of mRNA vaccines for COVID-19 is making steady progress, it appears that we are always a step too late as we are bombarded with new mutations of SARS-CoV-2. However, if we manage to establish a treatment that can prevent the aggravation of COVID-19 symptoms, regardless of the variant, by treating individuals with vaccine breakthrough infections while they only have mild to moderate symptoms, then observation from home may become possible [42]. Even if the number of cases rises, the potential for placing a strain on hospitals will, thus, remain low.

Since IVIG preparations are blood-derived preparations, the collection of blood samples is necessary, and the limitations this places on mass production cannot be ignored. However, treating patients with mild to moderate symptoms using a small amount of IVIG preparation will result in shortening the necessary period to cease business activities. A meta-analysis to investigate the efficacy of IVIG treatment in patients with COVID-19 has already been performed [43]. In the critical subgroup, IVIG was unable to reduce mortality compared with the control group; however, there was no significant difference in outcomes between the severe and non-severe subgroups. The strength of our study is that it dramatically supports the usefulness of IVIG for suppressing the exacerbation of COVID-19 in mild to moderate cases, using blood obtained from donors in the current pandemic period.

## 5. Conclusions

IVIG may contribute to therapy for COVID-19, including for cases caused by SARS-CoV-2 variants, since IVIG binds not only to the spike proteins of the virus, but also to human ACE2/TMPRSS2.

## Figures and Tables

**Figure 1 microorganisms-11-00471-f001:**
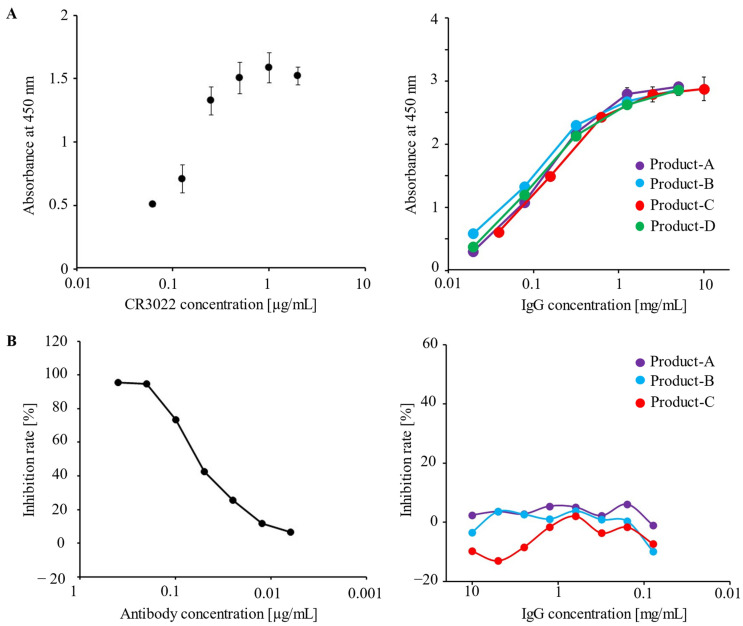
(**A**) The antigen-binding capacities of the IVIG preparations to the SARS-CoV-2 spike protein were measured by an indirect ELISA. The ELISA was conducted in triplicate, and the data are presented as the average ± standard deviation. ELISA binding curves of anti-SARS-CoV-2 spike antibody (CR3022; abcam) (left panel) or immunoglobulin preparations (right panel) to a full-length SARS-CoV2 spike protein. The experiment was conducted in triplicate, and the data are presented as the mean ± standard deviation. (**B**) The IgG antibodies against the receptor binding domain (RBD) on the SARS-CoV-2 spike protein in IVIG preparations were measured using the SARS-CoV-2 anti-RBD antibody profiling kit, and the average data of the RBD-ACE2 binding inhibition rate were measured in duplicate: a neutralizing antibody (positive control, left panel) and IVIG preparations (test samples, right panel). The experiment was conducted in duplicate, and the data are presented as the mean.

**Figure 2 microorganisms-11-00471-f002:**
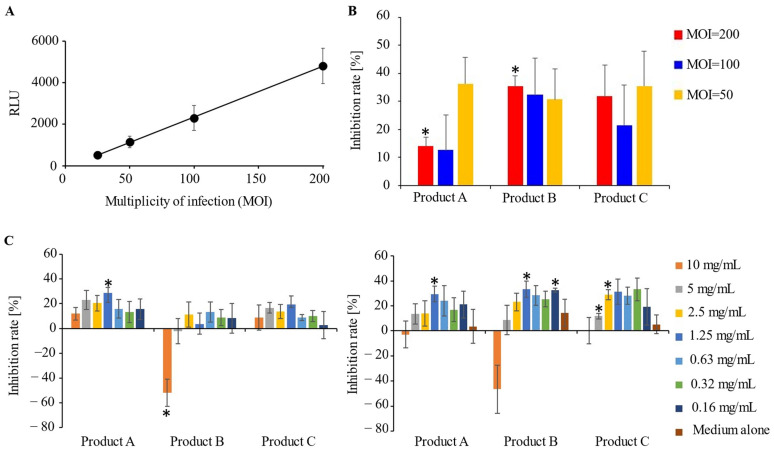
(**A**) Dose dependency of the SARS-CoV-2 pseudovirus in the pseudovirus entry assay. (**B**) Inhibition of infection by IVIG treatment (10 mg/mL) of ACE2/TMPRESS2-expressing HEK293T cells treated with IVIG preparations at 3 different MOIs. (**C**) The inhibitory effect by IVIG preparations was evaluated relative to RLU in the control group (MOI = 100, no treatment with IVIG preparation): IVIG treatment (0.16–10 mg/mL) of SARS-CoV-2 pseudovirus (left panel) and of SARS-CoV-2 pseudovirus + HEK293T cells (right panel). In (**A**–**C**), all assays were conducted in three independent experiments in sextuplicate, and the data are presented as the mean ± standard errors (* *p* < 0.05) (Student’s *t*-test).

## Data Availability

Not applicable.

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
