# Peer review of "Cross-Reactivity of Antibodies in Intravenous Immunoglobulin Preparation for Protection against SARS-CoV-2"

_microorganisms, 2023, doi:10.3390/microorganisms11020471_

Round 1

Reviewer 1 Report

Journal   micoorganisms

Article Cross-reactivity of antibodies in the intravenous-immuno-2 globulin preparation for the protection against SARS-CoV-2

This manuscript covers an important topic which is the cross reactivity in ivig for the protections of covid 19 .The topic is very interesting .However ,some points need to be addressed

Abstract

The methods and results should be clearer.

Introduction

59.. Our 59 study provides insight into IVIG therapy for COVID-19 including SARS-CoV-2 variants.

This sentence is a repetition. So .it will be better to delete it

Discussion

The strength and limitations of the study should be added

More relevant studies need to be added art the discussion section

The mechanism of action of ivig in this regard should be clear

It will be better to add information about the antiviral aspects of ivig and how it proved efficacy with other viral infection

Author Response

Authors' Responses to Reviewer's Comments (Reviewer 1)

1. Abstract

The methods and results should be clearer.

Response: The following text has been added to the Abstract to improve readers’ understanding:

“…using a SARS-CoV-2 pseudovirus assay” (Line 20) and “In contrast, antibody-dependent enhancement of viral entry was confirmed when SARS-CoV-2 pseudovirus was treated with some products at an IgG concentration of 10 mg/mL.” in Lines 24-26 in the revised manuscript.

  1. Introduction

59.. Our 59 study provides insight into IVIG therapy for COVID-19 including SARS-

CoV-2 variants.

This sentence is a repetition. So .it will be better to delete it

Response: As suggested, we have deleted this text.

  1. Discussion

The strength and limitations of the study should be added

Response: As suggested, we have now added the following text concerning the strengths of the study:

“The strength of our study is that it dramatically supports the usefulness of IVIG for suppressing exacerbation of COVID-19 in mild to moderate cases using blood obtained from donors in the current pandemic period” in Lines 283-286 in the revised manuscript.

In addition, we added the following text concerning the limitations of the study:

“Therefore, the effectiveness of IVIG using Japanese blood in the pre-pandemic term may be much lower than what was fundamentally intended.” in Lines 266-268 in the revised manuscript.

More relevant studies need to be added to the discussion section

Response: We agree with the reviewer’s comments and have now added a number of recent studies to the Discussion section (Lines 221-227; References#34-36). COVID-19 patients often complain of long-term sequelae in various organs, dubbed post-COVID-19 syndrome or long COVID. Patients with post-COVID syndrome demonstrate increased levels of inflammatory cytokines via the persistent activation of mononuclear phagocytes (e.g. macrophages, plasmacytoid dendritic cells). Treatment with IVIG has been applied to several inflammatory disorders, as IVIG confers anti-inflammatory and immuno-modulatory effects. Thus, IVIG may also be expected to contribute to the prevention of post-COVID syndrome by promoting convergence of the inflammatory response in COVID-19. In addition, a meta-analysis investigated the efficacy of IVIG treatment in patients with COVID-19. In the critical subgroup, IVIG was unable to reduce the mortality compared with the control group; however, there was no significant difference in outcomes between the severe and non-severe subgroups.

The mechanism of action of ivig in this regard should be clear. It will be better to add information about the antiviral aspects of ivig and how it proved efficacy with other viral infection.

Response: As suggested, we have now added the following text “It has recently been shown that COVID-19 severity is associated with higher ADCC activity and lower ADCP activity [33], which may suggest the involvement of another preventive mechanism of IVIG by inhibiting excess inflammation through its nonspecific binding to the FcRs of macrophages, in addition to the neutralizing effect of IVIG towards SARS-CoV-2 and the antibody-dependent cell-mediated cytotoxicity to them via FcRs” in Lines 216-221 in the revised manuscript.

Reviewer 2 Report

Osaka et al presented their results on human IVIG antibodies cross-reacting with SARS-CoV-2. This is a pre-clinical study using IVIG suspensions of healthy donors (received prior COVID-19 pandemic) which were tested on cell lines cultured to express ACE2/TMPRSS2 and introduced a preparation of SARS-CoV-2 spike protein-pseudotyped lentvirus. They found that IVIG prevent SARS-CoV-2 pseudovirus even at low antibody concentrations. This effect was attributed more to the blocking of cell surface molecules than to that of blocking SARS-CoV-2 spike protein. Their results are promising for the use of IVIG in the management of COVID-19.

This is a very interesting paper. The manuscript is well written. Methods are well justified, and the results well explained.

I believe that the paper should be accepted for publication.

I would like to make only some minor observations for the authors to consider.

i)                    Please provide a structured abstract.

ii)                  Introduction, line 39. Please specify that vaccines are used for prophylaxis and mAbs for therapy (seems like both are used for prophylaxis and therapy)

iii)                Line 55. I believe that there are more studies on IVIG and COVID you could cite. It would be useful for the reader to add some sentences on discussion regarding the results of these studies (and how they compare with your results)

Author Response

Authors' Responses to Reviewer's Comments (Reviewer 2)

Comments and Suggestions for Authors

Osaka et al presented their results on human IVIG antibodies cross-reacting with SARS-CoV-2. This is a pre-clinical study using IVIG suspensions of healthy donors (received prior COVID-19 pandemic) which were tested on cell lines cultured to express ACE2/TMPRSS2 and introduced a preparation of SARS-CoV-2 spike protein-pseudotyped lentvirus. They found that IVIG prevent SARS-CoV-2 pseudovirus even at low antibody concentrations.

This effect was attributed more to the blocking of cell surface molecules than to that of blocking SARS-CoV-2 spike protein. Their results are promising for the use of IVIG in the management of COVID-19.

This is a very interesting paper. The manuscript is well written. Methods are well justified, and the results well explained.

I believe that the paper should be accepted for publication.

Response: We are very pleased to note the favorable comments of the reviewer.

Please provide a structured abstract.

Response: While we are happy to provide a structured abstract if requested, a structured abstract does not seem to follow the journal’s submission guidelines, so we have not done so.

Introduction, line 39. Please specify that vaccines are used for prophylaxis and mAbs for therapy (seems like both are used for prophylaxis and therapy).

Response: We agree with the comment and have now revised the text as follows:

“At present, a vaccine-induced humoral-immune response and passive immunization with convalescent serum of COVID-19 patients and human monoclonal antibodies are used worldwide as effective prophylaxis and therapy against SARS-CoV-2 infection.” (Lines 42-45).

 Line 55. I believe that there are more studies on IVIG and COVID you could cite. It would be useful for the reader to add some sentences on discussion regarding the results of these studies (and how they compare with your results)

Response:  We have cited nine more representative studies (Reference number 15-19, 28-30, 43) and added the following text as suggested:

“A meta-analysis to investigate the efficacy of IVIG treatment in patients with COVID-19 has been already performed [43]  .In the critical subgroup, IVIG was unable to reduce the mortality compared with the control group; however, there was no significant difference in outcomes between the severe and non-severe subgroups. The strength of our study is that it dramatically supports the usefulness of IVIG for suppressing exacerbation of COVID-19 in mild to moderate cases using blood obtained from donors in the current pandemic period.” (Lines 280-286). 

Reviewer 3 Report

The paper submitted for review shows an interesting aspect in clinical practice such as the cross-reactivity of antibodies in the intravenous-immunoglobulin preparation for the protection against SARS-CoV-2.

The article is adequately structured, but I find the discussion section unclear, and above all it should make clearer the clinical implications of the authors' proposal. I believe that with the modification of the mentioned section, commenting more extensively and clearly what the clinical implications could be, the article could be sent for re-evaluation by the journal.

Author Response

Authors' Responses to Reviewer's Comments (Reviewer 3)

Comments and Suggestions for Authors

The paper submitted for review shows an interesting aspect in clinical practice such as the cross-reactivity of antibodies in the intravenous-immunoglobulin preparation for the protection against SARS-CoV-2.

Response: We found the reviewer 3’s comments extremely helpful and revised the manuscript accordingly.

The article is adequately structured, but I find the discussion section unclear, and above allit should make clearer the clinical implications of the authors' proposal.

Response: To make the Discussion section clearer, the following text has been moved from the Results to the Discussion: “In the family […] involved in viral infection.” in Lines 231-256 in the revised manuscript. In addition, the clinical implications have been described in the 2nd and 3rd paragraphs of the Discussion.

I believe that with the modification of the mentioned section, commenting more extensively

and clearly what the clinical implications could be, the article could be sent for re-evaluation

by the journal.

Response: To make our clinical implications of our proposal clear, we added the following text to the revised manuscript: “COVID-19 patients often complain of…or long COVID [34]." (Lines 221-223) and “The strength of our study is that it dramatically supports the usefulness of IVIG for suppressing exacerbation of COVID-19 in mild to moderate cases using blood obtained from donors in the current pandemic period.”(Lines 283-286).

Round 2

Reviewer 1 Report

The manuscript improved greatly and all the required modifications were done .

Author Response

Reviewer 1's Comment: The manuscript improved greatly and all the required modifications were done .

Response: : We are very pleased to note the favorable comments of the reviewer.

The English has been checked carefully and corrected by ‘MDPI Language Editing Services’.

We trust that the revised manuscript is acceptable for publication in ‘Microorganisms’.

Reviewer 3 Report

The article sent for review with the modifications suggested by the reviewers has improved considerably, responding to the suggested modifications, considering it of interest for publication in the journal. 

Author Response

Authors' Responses to Reviewer's Comments (Reviewer Comments and Suggestions for Authors)

The article sent for review with the modifications suggested by the reviewers has improved considerably, responding to the suggested modifications, considering it of interest for publication in the journal.  

Response: We are very pleased to note the favorable comments of the reviewer.

The English has been checked carefully and corrected by ‘MDPI Language Editing Services’.

We trust that the revised manuscript is acceptable for publication in ‘Microorganisms
